# Unconventional SCC*mec* types and low prevalence of the Panton-Valentine Leukocidin exotoxin in South African blood culture *Staphylococcus aureus* surveillance isolates, 2013-2016

Ashika Singh-Moodley[1,2]*, Wilhelmina Strasheim[1], Ruth Mogokotleng[1], Husna Ismail[1], Olga Perovic[1,2]

**1** Centre for Healthcare-Associated Infections, Antimicrobial Resistance and Mycoses, National Institute for Communicable Diseases, a Division of the National Health Laboratory Service, Johannesburg, South Africa, **2** Department of Clinical Microbiology and Infectious Diseases, School of Pathology, Faculty of Health Sciences, University of Witwatersrand, Johannesburg, South Africa

* ashikas@nicd.ac.za

**Data Availability Statement:** All relevant data are within the paper.

## Abstract

*Staphylococcus aureus* is a healthcare-associated pathogen that can harbour multiple antimicrobial resistance determinants and express multiple virulence factors e.g. Panton-Valentine Leukocidin (PVL). Unknown staphylococcal cassette chromosome *mec* (SCC*mec*) typing patterns were previously observed among 11% (n = 52) of methicillin-resistant *S. aureus* (MRSA) isolates; we further investigated these as well as the proportion of PVL, encoded by *luk*S/F-PV, in 761 *S. aureus* isolates from patients with a diagnosis of pneumonia/lower respiratory tract, skin/soft tissue, bone and joint infection. *S. aureus* isolates from blood culture were identified and antimicrobial susceptibility testing was performed using automated systems. Conventional PCR assays were used to identify the *ccr* and *mec* gene complexes in *mec*A-positive isolates with an unknown SCC*mec* type and screen for *luk*S/F-PV. Epidemiological data was used to classify isolates as healthcare- or community-associated infections. Antimicrobial susceptibility profiles according to SCC*mec* type and PVL were reported. Of the unknown SCC*mec* types, isolates were interpreted as type I-like (86%, 38/44), type II-like (9%, 4/44) and type III-like (5%, 2/44). Eight isolates did not produce definitive results. Of all MRSA isolates, majority were multidrug-resistant as indicated by their non-susceptibility to most antimicrobial agents; 92% were healthcare-associated. PVL was seen in 14% of the isolates (MRSA: 25%, MSSA: 75%); 56% were classified as healthcare-associated infection. The SCC*mec* typing method did not definitively classify all unknown isolates into clearly defined types. It showed that majority of these isolates were not the conventional types; untypeable elements appeared to be composite SCC*mec* elements, consisting of multiple *ccr* gene complexes. Majority of the MRSA isolates were non-susceptible to most antibiotics indicating that multiple resistance genes are present in our population. Furthermore, the proportion of PVL was low and more prevalent in MSSA.

**Funding:** This work was funded by the National Institute for Communicable Diseases and received no specific grant from any funding agency.

**Competing interests:** The authors have declared that no competing interests exist.

## Introduction

*Staphylococcus aureus* is a leading cause of community- and healthcare-associated infections. It is a pathogenic organism capable of causing a range of infections. This is due to its ability to express multiple virulence factors and harbour multiple antimicrobial resistance determinants [1, 2]. The Panton-Valentine Leukocidin (PVL) exotoxin is a pore-forming cytotoxic exoprotein encoded by the bicomponent *luk*S-PV and *luk*F-PV genes, which are carried on a bacteriophage [3]. PVL targets cells of the human immune system and causes cell death. Isolates of *S. aureus* harbouring PVL are often due to community-associated infections [4] and have been linked to more severe clinical manifestations such as necrotising pneumonia, severe bone and joint infections and skin and soft tissue infections often requiring surgical drainage [4, 5].

The PVL exotoxin is not produced by all *S. aureus* strains, but have been detected in both methicillin-susceptible *S. aureus* (MSSA) and methicillin-resistant *S. aureus* (MRSA) strains [3]. The clinical presentation of PVL-positive MSSA and MRSA is very similar. It has also been shown that sub-inhibitory concentrations of beta-lactams increases PVL toxin production *in vitro* and that other antimicrobials that inhibit protein synthesis (e.g. rifampicin, clindamycin and linezolid) should be considered during treatment [6].

Methicillin-resistance is harboured on a mobile genetic element called the staphylococcal chromosome cassette *mec* (SCC*mec*). This element is genetically diverse and many types, subtypes and variants have been reported [7]. The cassette consists of three main structural components, namely: i) the cassette chromosome recombinase (*ccr*) gene complex, ii) the *mec* gene complex and iii) the joining (J) regions [8, 9]. The *ccr* gene complex encodes for site-specific recombinases that are responsible for the excision and integration of the SCC*mec* element into the staphylococcal chromosome [2, 7, 9]. The *ccr* gene complex provides the SCC*mec* element with mobility and consequently facilitates its transfer to other staphylococcal species [9]. The *mec* complex is responsible for conferring methicillin resistance. The *mec* gene complex consists of: i) a *mec*A/C gene, ii) its regulatory genes, the *mec*R1 and the *mec*I genes encoding for a signal transducer protein and repressor protein respectively and iii) various insertion sequences [7, 10]. A combination of the *ccr* gene complex with the *mec* gene class is used to assign the specific type of SCC*mec* element. Currently, thirteen SCC*mec* types (I-XIII) based on complete sequence data have been defined in MRSA [2, 11–13]; International Working Group on the Staphylococcal Cassette Chromosome elements (IWG-SCC) (2015) Available online: http://www.sccmec.org).

Typing of the SCC*mec* element is of epidemilogical importance to understand the evolution of MRSA. There are several methods (i.e. restriction enzyme digestion, PCR assays) available to determine the SCC*mec* type. The International Working Group for SCC*mec* elements currently recommends the method by Kondo and colleagues (2007) [12]. The only drawback of this assay is that it consists of six multiplex-PCR assays. However, the first two assays are generally sufficient to classify the majority of SCC*mec* elements [9]. The assay by Kondo and colleagues (2007) can easily detect novel *ccr* and *mec* gene complex combinations. [2, 12].

Healthcare-associated MRSA infections have previously been associated with SCC*mec* types I, II or III whereas community-acquired MRSA infections are linked to smaller SCC*mec* types IV, V, VI or VII [14] but epidemiological data is required to make this conclusion. A more recent report however, states that the traditional classification of healthcare- and community-associated MRSA is no longer appropriate because there is a notable overlap of identical clones between these two groups [15]. Previous work characterised the isolates in this study to determine the circulating SCC*mec* types [16]. However, a number of isolates produced an unknown SCC*mec* type as indicated by unidentified banding patterns. This study aimed to resolve unidentified SCC*mec* types using an alternative method. In addition, selected *S. aureus*

isolates have been screened for the presence of the *luk*S/F-PV gene. Epidemiological data was used to classify isolates as healthcare- and community-associated and specific antimicrobial susceptibility profiles according to SCC*mec* type and PVL were reported.

## Methods

### Case definition

A case of *S. aureus* bacteraemia was defined as the isolation of *S. aureus* from a blood culture. Isolates formed part of the GERMS-SA enhanced antimicrobial resistance surveillance study from January 2013 to January 2016. *S. aureus* isolates were from five sentinel centres in South Africa and sites represented six large academic hospitals from the Gauteng and the Western Cape provinces. A 21-day exclusion period was applied to avoid duplicate isolates of the organism from the same patient. In addition, the following clinical and epidemiological information were collected for each case: date of admission, previous hospitalisation or treatment and diagnosis. This information was used to define healthcare- and community-associated *S. aureus* infections. A case of community-associated bacteraemia was defined as a patient with *S. aureus* isolated from a blood culture specimen ≤ 48 hours of admission to a hospital and no contact within one year prior to the current episode of *S. aureus* infection with a healthcare facility (including prior surgery, dialysis and admission to a long term care facility). Healthcare-associated *S. aureus* infection was defined as a patient with *S. aureus* isolated >48 hours after admission or with any prior healthcare contact. The patient's clinical diagnosis was used to select isolates for *luk*S/F-PV (i.e. PVL) screening.

### Bacterial isolates and phenotypic assays

Isolates were submitted on Dorset transport media (Diagnostic Media Products, National Health Laboratory Service, South Africa) to the National Institute for Communicable Diseases (NICD), Johannesburg, South Africa. Each isolate was plated onto a 5% blood agar plate (Diagnostic Media Products, National Health Laboratory Service, South Africa) followed by organism identification and antimicrobial susceptibility testing using automated systems [(VITEK II system (bioMèrieux, Marcy-l'Etoile, France)/MALDI-TOF MS (Microflex, Bruker Daltonics, MA, USA) and MicroScan Walkaway system (Siemens, Sacramento, CA, USA), (Gram-positive panel PM33)], respectively. An isolate was phenotypically non-susceptible if it had an oxacillin MIC of >2 and a positive cefoxitin screening result. Interpretation of susceptibility was performed according to the Clinical and Laboratory Standards Institute (CLSI) guidelines [17].

### Genotypic assays

Total genomic DNA was extracted using a crude boiling method and used in the genotypic assays. Fig 1 shows how isolates were selected for the genotypic assays. Prior work involved the screening for methicillin-resistance determinants, *mec*A and *mec*C on 484 phenotypically confirmed MRSA isolates. These isolates were typed by multiplex PCR to determine the circulating SCC*mec* types I to VI as described previously [18].

### SCC*mec* typing of unidentified banding patterns

An alternate SCC*mec* typing method was carried out by performing two conventional multiplex-PCR assays (i.e. *ccr* gene complex multiplex PCR assay detecting *ccr* gene complex 1 to 5 and the *mec* gene complex multiplex PCR assay detecting *mec* class A, B and C) using the G-Storm thermal cycler (Somerton Biotechnology Centre, Somerton, UK), the Qiagen

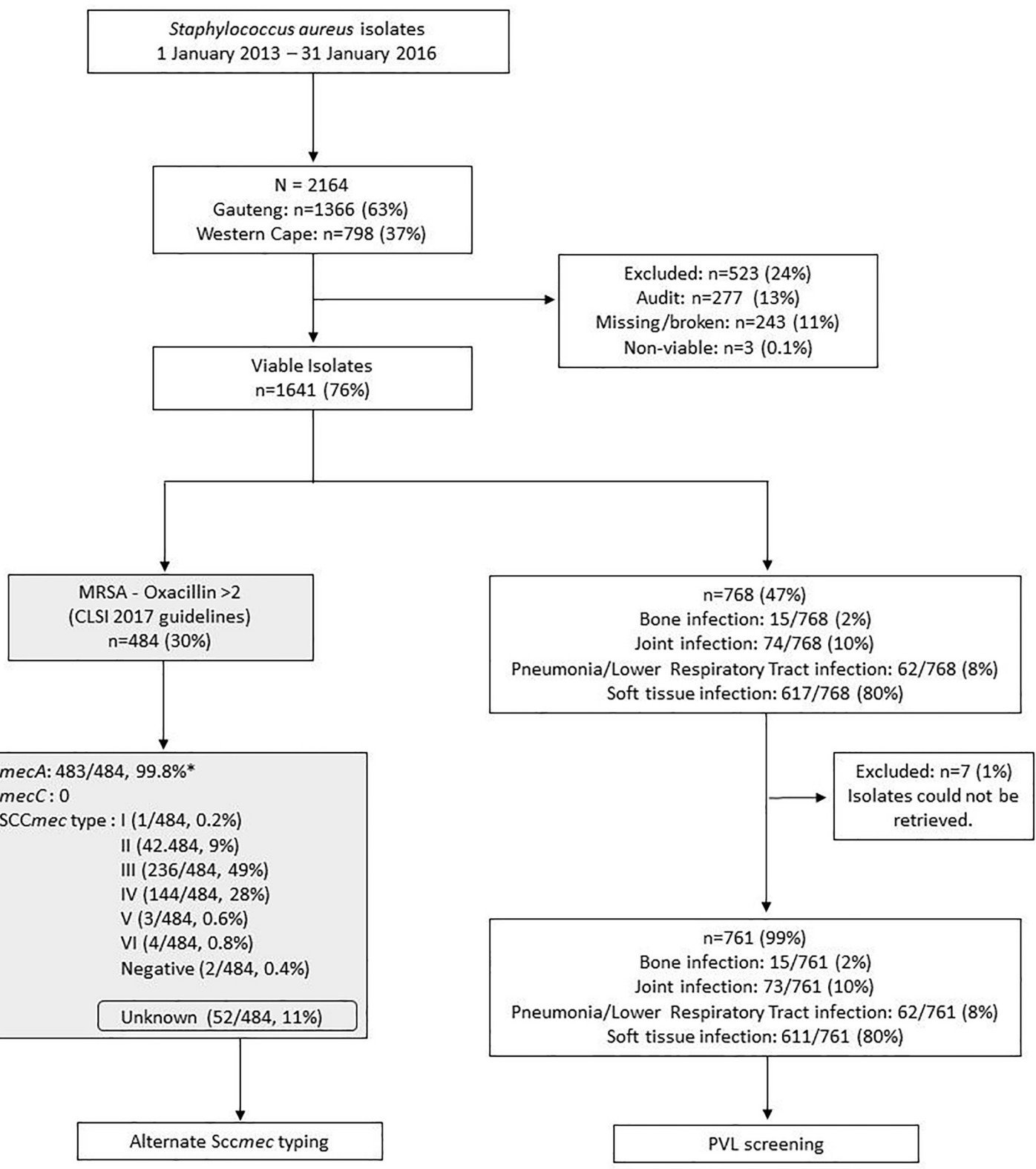

**Fig 1. Flowchart showing sample selection.** The grey boxes show results that have been published previously [18]. Audit cases were defined as those cases that were identified according to the public healthcare sector Corporate Data Warehouse (CDW) (which houses records of patient details and laboratory results) but not received for processing in the laboratory. * One isolate did not contain *mec*A or *mec*C and did not harbour a SCC*mec* element although it was phenotypically non-susceptible (oxacillin >2, cefoxitin screen positive). This is possible as explained previously [19] possibly due to excision of the cassette (SCC*mec*) and/or *mec*A/C gene drop-out.

Multiplex PCR kit (Qiagen, Nordrhein-Westfalen, Germany) and previously published primers [12]. The results of the *ccr* gene complex and the *mec* gene complex multiplex PCR assays were subsequently combined to assign a specific SCC*mec* type.

### PCR screening for the *luk*S/F-PV gene

The presence of the *luk*S/F-PV gene was determined in MRSA and MSSA isolates from patients with a diagnosis of pneumonia or a lower respiratory tract infection, skin and soft tissue infection, bone infection and joint infection. Isolates were screened by a conventional PCR assay using the G-Storm thermal cycler, the Qiagen Multiplex PCR kit and previously published primers [20].

### Statistical analysis

Descriptive analyses were performed using Stata version 14 (StataCorp LP, College Station, Texas, USA). Categorical data were summarised using absolute frequencies and percentages.

### Ethical approval

Ethical clearance was obtained from the University of the Witwatersrand Human Research Committee (Protocol number M10464).

## Results

### SCC*mec* typing of unidentified banding patterns

Only 51 isolates were tested with the *ccr* and the *mec* gene complex multiplex PCR assays, since a single isolate could not be retrieved from storage. A definite SCC*mec* type could not be determined for a three isolates, since all three of these isolates harboured two *ccr* gene complexes (*ccr* gene complex 1 and *ccr* gene complex 5) but was negative for all *mec* gene classes tested. A single isolate produced an indeterminate result and harboured either SCC*mec* type V or VII, since the assay by Kondo *et al*., 2007 [12] cannot distinguish between *mec* gene complex class C1 and C2. A further 44 isolates produced evidence of composite SCC*mec* elements as they harboured two *ccr* gene complexes and a single *mec* gene complex. These isolates were further interpreted as SCC*mec* type I-like (1B plus an additional *ccr* gene complex class 5) (86%, 38/44), SCC*mec* type II-like (2A plus an additional *ccr* gene complex class 5) (9%, 4/44) and SCC*mec* type III-like (3A plus an additional *ccr* gene complex class 5) (5%, 2/44). The SCC*mec* element type could clearly be assigned for two isolates only based on the *ccr* and *mec* gene complex combinations; these were SCC*mec* type II and IV. A new SCC*mec* combination (*ccr* gene complex class 5 and *mec* gene complex class A) was detected in a single isolate. Conclusive SCC*mec* types making use of both typing methods are seen in Fig 2.

Epidemiological data were available for 463 of the 484 isolates that could be classified into healthcare- and community-associated *S. aureus* infection. Majority of these were healthcare-associated infections (92%, 428/463) followed by community-associated infections (8%, 35/463). Table 1 shows a breakdown of the SCC*mec* element types classified into healthcare- and community-associated infections.

### Antimicrobial susceptibility testing and SCC*mec* types

Table 2 shows AST data classified by the predominant SCC*mec* element types. Majority of the isolates were multidrug-resistant as indicated by their non-susceptibility to most of the antimicrobial agents. The p-value was calculated between susceptible (S) and the non-susceptible (I and R) isolates collectively and was significant for all antimicrobial agents.

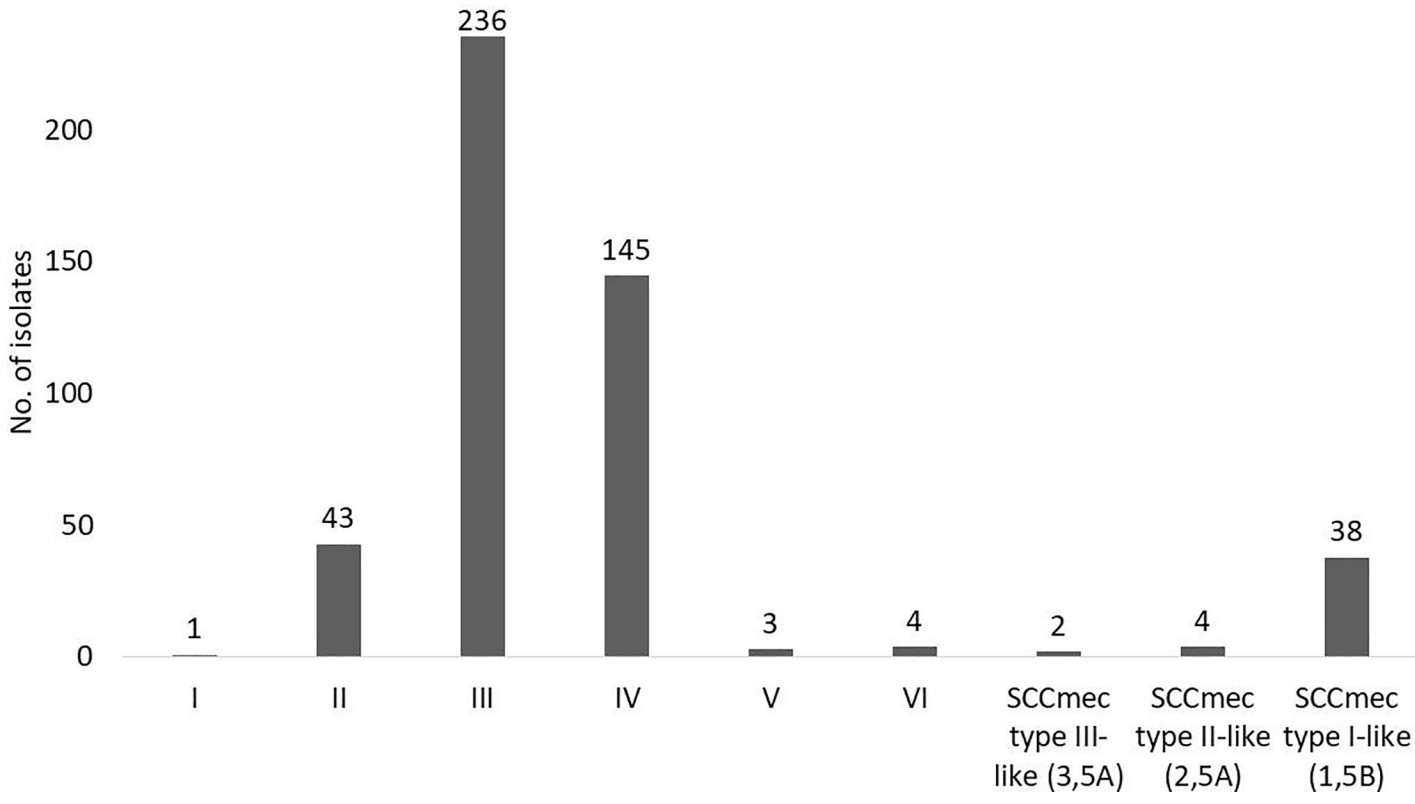

**Fig 2. Final SCC*mec* type distribution among MRSA isolates 2013–2016.** A total of 476 isolates are represented in this figure. One isolate could not be retrieved for testing, two isolates produced no SCC*mec* type, the *mec* class could not be identified for three isolates, one isolate could not be differentiated between types V or VII, and one isolate produced a new combination.

**Table 1. Final SCC*mec* type distribution among MRSA isolates from 2013 to 2016 and its association between origin of the infection (i.e. healthcare- or community associated).**

|  | Healthcare-associated n/463 (%) | Community-associated n/463 (%) |
|---|---|---|
| SCC*mec* type I | 0 | 1 (0.2) |
| SCC*mec* type II | 43 (9) | 0 |
| SCC*mec* type III | 218 (47) | 9 (2) |
| SCC*mec* type IV | 116 (25) | 19 (4) |
| SCC*mec* type V | 2 (0.4) | 0 |
| SCC*mec* type VI | 3 (0.7) | 1 (0.2) |
| SCC*mec* type V or VII | 1 (0.2) | 0 |
| SCC*mec* type III-like | 2 (0.4) | 0 |
| SCC*mec* type II-like | 4 (0.9) | 0 |
| SCC*mec* type I-like | 33 (7) | 4 (0.9) |
| New combination | 1 (0.2) | 0 |

A single isolate could not be retrieved for testing, three isolates in which the *mec* gene class could not be detected were classified as healthcare-associated (n = 2) and community-associated (n = 1) and two isolates that produced no SCC*mec* banding pattern were classified as healthcare-associated.

**Table 2. Susceptibility of antimicrobial agents in MRSA isolates.**

| Antimicrobial agents MIC range (μg/mL) | Number of isolates for which MIC value was: | | | | p-value* |
|---|---|---|---|---|---|
| | SCC*mec* type II (N = 43) | SCC*mec* type III (N = 236) | SCC*mec* type IV (N = 145) | SCC*mec* type I-like (1,5B) (N = 38) | |
| Ciprofloxacin | | | | | |
| ≤ 1 (S) | - | 3 | 13 | 32 | <0.001 |
| 2 (I) | - | - | 1 | 2 | |
| >2 (I) | 43 | 233 | 131 | 4 | |
| ≥4 (R) | - | - | - | - | |
| Clindamycin | | | | | |
| ≤0.5 (S) | 4 | 212 | 139 | 36 | <0.001 |
| 1–2 (I) | - | 3 | - | - | |
| >2 (I) | 39 | 21 | 6 | 2 | |
| ≥4 (R) | - | - | - | - | |
| Erythromycin | | | | | |
| ≤0.5 (S) | - | 1 | 61 | - | <0.001 |
| 1-4(I) | - | - | 5 | 1 | |
| >4 (I) | 43 | 233 | 79 | 37 | |
| ≥8 (R) | - | 1 | - | - | |
| Gentamicin | | | | | |
| ≤4 (S) | 35 | 3 | 31 | 3 | <0.001 |
| 8 (I) | 1 | - | 4 | 8 | |
| >8 (I) | 7 | 232 | 110 | - | |
| ≥16 (R) | - | - | - | - | |
| Rifampicin | | | | | |
| ≤ 1 (S) | 39 | 219 | 32 | 29 | <0.001 |
| 2 (I) | - | 3 | - | - | |
| >2 (I) | 3 | 14 | 112 | 1 | |
| ≥4 (R) | - | - | 1 | - | |
| Tetracycline | | | | | |
| ≤4 (S) | 39 | 6 | 31 | 34 | <0.001 |
| 8 (I) | 1 | 1 | - | 2 | |
| >8 (I) | 3 | 228 | 114 | 2 | |
| ≥16 (R) | - | - | - | - | |
| Trimethoprim/sulfamethoxazole | | | | | |
| ≤2/38 (S) | 38 | 9 | 28 | 37 | <0.001 |
| ≥4/76 | 4 | 225 | 106 | 1 | |

MIC: Minimum inhibitory concentration.

Interpretation of results was done according to CLSI guidelines [17]; S: susceptible, I: Intermediate resistance, R: Resistant.

*The p-value was calculated between susceptible (S) and the non-susceptible (I and R) isolates collectively.

Multidrug resistance was defined as non-susceptibility to at least one antimicrobial agents in three or more categories [21].

## PCR screening for the *luk*S/F-PV gene

Of a total of 1641 isolates, 768 (47%) isolates were selected for screening for the *luk*S/F-PV gene based on the type of infection. Isolates from patients diagnosed with skin and soft tissue infection (80%, 617/768), pneumonia/lower respiratory tract infection (8%, 62/768), bone infection (2%, 15/768) or joint infection (10%, 74/768) were selected.

**Table 3. Presence of SCC*mec* types and prevalence of the Panton-Valentine Leukocidin gene in MRSA isolates (n = 27).**

| SCCmec type | Type of infection | PVL-positive strains (n, %) |
|---|---|---|
| II | Skin and soft tissue infection | 1 (4) |
| III | Skin and soft tissue infection | 8 (30) |
| | Joint Infection | 2 (7) |
| | Lower respiratory tract infection | 1 (4) |
| IV | Skin and soft tissue infection | 8 (30) |
| | Joint Infection | 1 (4) |
| | Bone Infection | 1 (4) |
| | Lower respiratory tract infection | 3 (10) |
| V | Joint Infection | 1 (4) |
| VI | Joint Infection | 1 (4) |

Seven isolates could not be retrieved and were therefore not tested for the presence of *luk*S/F-PV. The presence of the PVL toxin was seen in 14% (108/761) of the isolates tested; 25% (27/108) in MRSA and 75% (81/108) in MSSA. Majority of PVL-positive *S. aureus* cases were from the Gauteng (75%, 81/108) followed by the Western Cape province (25%, 27/108). Majority of these infections were from patients with skin and soft tissue infection (70%, 75/108), followed by pneumonia/lower respiratory tract infection (21%, 23/108), joint infection (8%, 9/108) and bone infection (1%, 1/108) ($p < 0.0001$).

Eleven (41%) and 13 (48%) of 27 MRSA isolates that were PVL-positive were associated with SCC*mec* types III and IV respectively with just one isolate each for types I, V and VI (Table 3).

Table 4 shows the antimicrobial susceptibility profiles for PVL-positive *S. aureus* isolates according to patient's diagnosis.

Epidemiological data for five isolates were missing. Of 103 PVL-positive isolates, 56% (58/103) were classified as healthcare-associated *S. aureus* infection followed by 44% (45/103) that were community-associated infection.

## Discussion

In the current study, MRSA accounted for 30% of the total staphylococcal bacteraemia infections. SCC*mec* element typing was performed on these MRSA isolates, but 11% could not be typed by the multiplex PCR assay described by Milheiriço and colleagues (2007) [22]. An alternate SCC*mec* typing method was employed to further classify untypeable SCC*mec* elements based on *ccr* and *mec* gene complex combinations. In addition, the association between antimicrobial susceptibility profiles, specific SCC*mec* types, the epidemiological point of acquisition (i.e. healthcare-associated vs. community-acquired infection) and the presence of the PVL toxin were reported.

The study showed that the majority of untypeable isolates harboured more than one *ccr* gene complex. This indicates the possibility of composite SCC*mec* elements circulating in MRSA strains in South Africa. Sequencing of the complete element would be required to determine if these elements represent a single SCC*mec* element with two *ccr* gene complexes or if the element consists of two separate integrated SCC*mec* elements [23]. It is also important to note that the method published by Kondo and colleagues (2007) [12] possesses some limitations. The *ccr* gene complex multiplex-PCR assay cannot detect *ccr* gene complex 6 to 9, hence SCC*mec* type X (*ccr* gene complex 7), SCC*mec* type XI (*ccr* gene complex 8), SCC*mec* XII (*ccr* gene complex 9) and SCC*mec* type XIII (*ccr* gene complex 9) would not have been detected.

**Table 4. Antimicrobial susceptibility profiles for PVL-positive *S. aureus* isolates according to patient's diagnosis.**

| Antimicrobial agents MIC range (μg/mL) | Patient's diagnosis | | | |
|---|---|---|---|---|
| | Skin and soft-tissue infections n = 75 | Pneumonia / Lower respiratory tract infection n = 23 | Bone infection n = 1 | Joint infection n = 9 |
| **Oxacillin resistant /cefoxitin screen positive** | 17 | 4 | 1 | 5 |
| **Trimethoprim/sulfamethoxazole** | | | | |
| ≤2/38 (S) | 57 | | | |
| >2.38 (R) | 1 | | | |
| 4/76 (R) | 2 | | | |
| >4/76 (R) | 15 | | | |
| **Clindamycin** | | | | |
| ≤0.25 (S) | | 23 | 1 | 8 |
| 0.5 (S) | | 0 | 0 | 1 |
| 1 (R) | | 0 | 0 | 0 |
| >2 (R) | | 0 | 0 | 0 |
| **Daptomycin** | | | | |
| ≤1 (S) | | | 1 | 9 |
| **Linezolid** | | | | |
| ≤1 (S) | | 15 | 0 | 3 |
| ≤2(S) | | 0 | 0 | |
| 2 (S) | | 8 | 1 | 6 |
| **Rifampicin** | | | | |
| ≤0.5 (S) | | 19 | | |
| ≤1 (S) | | 1 | | |
| 2 (I) | | 3 | | |
| >2 (R) | | 0 | | |

S–Susceptible, I–Intermediate resistance, R–Resistant

Shaded grey boxes indicate that the antimicrobial agents is not the recommended treatment of choice.

Furthermore, the *mec* gene complex multiplex-PCR assay cannot distinguish between *mec* class C1 and C2 and only detects the *mec* gene complex C resulting in not being able to distinguish between SCC*mec* type V (*mec* gene complex C2) and SCC*mec* type VII (*mec* gene complex C1). This assay also does not detect the *mec* gene complex E and therefore SCC*mec* type XI (*mec* gene complex E) would not be detected.

While this method did not definitively classify all the unknown isolates into clearly defined types, it showed that majority of these isolates were not the conventional SCC*mec* types. It should be noted that only two of the six multiplex-PCR assays were performed; however the first two assays are generally sufficient to classify the majority of SCC*mec* elements [9]. For this reason we performed only the two PCR assays. Types I, II and III are thought to be related to healthcare-associated infections and types IV, V and VI with community-associated infections [14]; this was the case for a large number of the isolates except for 116 SCC*mec* type IV isolates which were classified as healthcare-associated. A further few did not strictly behave as traditionally expected. A more recent report however, states that the traditional classification of healthcare-associated and community-associated MRSA is no longer appropriate because there is a notable overlap of identical clones between these two groups. This may confirm the limitation in defining healthcare-associated and community-associated infections based on SCC*mec* types and that traditionally considered community-associated isolates have the potential to establish themselves as healthcare-associated pathogens and vice versa. [15].

The results obtained demonstrate that majority of the *S. aureus* isolates displaying resistance to methicillin were non-susceptible to ciprofloxacin, erythromycin, gentamicin, rifampicin, tetracycline and trimethoprim/sulfamethoxazole while susceptible to clindamycin only, indicating that multiple resistance genes are present (Table 2). A previous 2010 Swiss study [24] showed some differences in their findings although the sample number was small (N = 78); 60% were resistant to ciprofloxacin compared to 90% in our findings, 33% were resistant to erythromycin compared to 87% in our study and 18% were resistant to gentamycin as compared to 83% in our study. Rifampicin and trimethoprim/sulfamethoxazole were resistant in none of the isolates from the Swiss study; however our findings showed that 30% and 75% were resistant respectively. Clindamycin displayed 15% resistance in our study and this was comparable to the Swiss study where 16% of the isolates were resistant. As in our study, erythromycin resistance was also quite high in 91.6% (n = 564) of the isolates in another study [25] but comparable to the Swiss study in another study (32.8%) [26]. This latter study looked at nasal colonisation with MRSA from the nasal secretion from children in day-care centres. Resistance to ampicillin was high (80%) but resistance to ciprofloxacin, clindamycin and tetracycline was low (7.1%, 7.1% and 4.3% respectively) [26]. The differences in the results between studies may be attributed to prescription practices which vary between hospitals and geographical locations.

A small American study (N = 60) showed that overall the healthcare-associated SCC*mec* type II isolates were resistant to more antimicrobials than the community-associated type IV isolates [27]. This was also seen in another small (N = 65) Polish study in 2013 [28]. The opposite was true in our study where type IV isolates were resistant to more antibiotics than the type II isolates. It should however be noted that in our study 116 SCC*mec* type IV isolates were classified as healthcare—associated infections and not community-associated infections.

Less than 50% of the isolates were screened for the presence of the PVL toxin as indicted by patient diagnosis; the presence of which was seen in 14% (n = 108/761) of the isolates tested; 25% (27/108) in MRSA and 75% (81/108) in MSSA. A 2012 Turkish study also showed a low proportion (2.2%, n = 10) of PVL-positive isolates, all of which were MSSA [4]. In the current study, 11 (41%) and 13 (48%) of 27 MRSA isolates that were PVL-positive were associated with SCC*mec* types III and IV respectively with just one isolate each for types I, V and VI. A previous study showed that 81.5% (22/27) PVL-positive isolates were associated with type IV elements but none of the type I, II or III elements detected were associated with the presence of the *luk*S/F-PV gene [29]. Another study that further characterised type IV isolates showed that these isolates were PVL-positive [30]. Isolates of *S. aureus* harbouring PVL are often because of community-associated infections [4]; however our findings showed that 56% were classified as healthcare-associated *S. aureus* infection followed by 44% that were community-associated infection. This was in contrast to a 2010 Swiss study where a high proportion (87%) of community-associated types IV and V among their PVL-positive strains was demonstrated [24]. A majority (48%) of the PVL-positive isolates were associated with SCC*mec* type IV (Table 3) followed by type III (41%) with only one each for types II, V and VI; this was consistent with the 2007 American study where 23 of 25 SCC*mec* type IV isolates were PVL-positive compared to the type II elements (n = 34) which did not harbour the gene [27], as well as with a 2013 Polish study where all the type IV isolates were PVL-positive (9/9) and none of the type II isolates (0/16) harboured the gene. In this latter study only three of 40 type III isolates were positive for PVL [28]. All of these differences highlight the evolving nature of *S. aureus* and variations in clones as a result of evolution is evident from one geographical region to another.

According to a recent publication [3] the recommended treatment of choice for PVL-positive *S. aureus* infections are as follows; skin and soft tissue infections: Trimethoprim / sulfamethoxazole, bone and joint infections: flucloxacillin combined with clindamycin for MSSA

and daptomycin combined with linezolid for MRSA, and pneumonia: clindamycin, linezolid or rifampicin. When we further analysed the antimicrobial susceptibility profiles of the PVL-positive isolates, we found that the treatment of choice as recommended by Saeed *et. al.*, [3] would be ideal for treating *S. aureus* skin and soft tissue infections and pneumonia as indicated by the high number of susceptible isolates to each of the antimicrobial agents. The ten PVL-positive isolates from patients with bone and joint infections were also susceptible to all antibiotics except for oxacillin. This is encouraging as it offers available treatment options.

This study aimed to resolve unidentified SCC*mec* types using an alternative method. While this method did not definitively classify all the unknown isolates into clearly defined types, it showed that majority of these isolates were not the conventional SCC*mec* types. Majority of the *S. aureus* isolates displaying resistance to methicillin were non-susceptible most antimicrobial agents indicating that multiple resistance genes are present in our population. The presence of PVL more prevalent in MSSA and was seen in small percentage of isolates indicating a low proportion of this exotoxin in our population.

## Acknowledgments

We thank Cheryl Hamman, Jenna Allen, Rubeina Badat, Naseema Bulbulia, Rosah Mobokachaba, Gloria Molaba and Marshagne Smith for assistance with the laboratory work and Boniwe Makwakwa for assistance with the database.

## Author Contributions

**Conceptualization:** Ashika Singh-Moodley, Olga Perovic.

**Formal analysis:** Ashika Singh-Moodley, Husna Ismail.

**Investigation:** Ashika Singh-Moodley, Olga Perovic.

**Methodology:** Ashika Singh-Moodley, Wilhelmina Strasheim, Ruth Mogokotleng.

**Project administration:** Ashika Singh-Moodley, Olga Perovic.

**Supervision:** Ashika Singh-Moodley, Olga Perovic.

**Writing – original draft:** Ashika Singh-Moodley, Wilhelmina Strasheim.

**Writing – review & editing:** Ashika Singh-Moodley, Wilhelmina Strasheim, Ruth Mogokotleng, Husna Ismail, Olga Perovic.

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
