## [Decision Letter · Decision Letter 0]

1 Oct 2019

PONE-D-19-23434

Investigation of virulence factors: SCCmec types and the Panton-Valentine Leukocidin exotoxin in South African blood culture Staphylococcus aureus surveillance isolates, 2013-2016.

PLOS ONE

Dear Dr Singh-Moodley

Thank you for submitting your manuscript to PLOS ONE. After careful consideration, we feel that it has merit but does not fully meet PLOS ONE’s publication criteria as it currently stands. Therefore, we invite you to submit a revised version of the manuscript that addresses the points raised during the review process. The major concerns of the Reviewers relate to the quality of the writing. I highly recommend thoroughly examining the grammar and spelling throughout your submission. 

We would appreciate receiving your revised manuscript by October 28th. To enhance the reproducibility of your results, we recommend that if applicable you deposit your laboratory protocols in protocols.io, where a protocol can be assigned its own identifier (DOI) such that it can be cited independently in the future. For instructions see: http://journals.plos.org/plosone/s/submission-guidelines#loc-laboratory-protocols

We look forward to receiving your revised manuscript.

Kind regards,

Davida S. Smyth, Ph.D.

Academic Editor

PLOS ONE

Journal Requirements:

2. In ethics statement in the manuscript and in the online submission form, please provide additional information about the patient records used in your retrospective study. Specifically, please ensure that you have discussed whether all data were fully anonymized before you accessed them and/or whether the IRB or ethics committee waived the requirement for informed consent. If patients provided informed written consent to have data from their medical records used in research, please include this information.

Additional Editor Comments (if provided):

Reviewers' comments:

Reviewer's Responses to Questions

**Comments to the Author**

1. Is the manuscript technically sound, and do the data support the conclusions?

Reviewer #1: No

Reviewer #2: Yes

2. Has the statistical analysis been performed appropriately and rigorously? 

Reviewer #1: N/A

Reviewer #2: Yes

3. Have the authors made all data underlying the findings in their manuscript fully available?

Reviewer #1: Yes

Reviewer #2: Yes

4. Is the manuscript presented in an intelligible fashion and written in standard English?

Reviewer #1: No

Reviewer #2: Yes

5. Review Comments to the Author

Reviewer #1: The authors report in this manuscript an analysis and findings with the S. aureus strains isolated from blood of patients between 2013 to 2016 in South Africa. Majority of these samples appear to come from the patients with skin and soft tissue infections. The authors have made an attempt to classify the S. aureus strains as MRSA or MSSA. The MRSA strains have been further assigned into different SCCmec complexes. Some isolates could not be assigned to a specific SCCmec complex using common approaches so the authors have used additional molecular approaches. The authors have also made an effort to detect the presence of lukS/F-PV gene among the S. aureus isolates and determine the resistance to various antibiotics for the MRSA isolates. Overall, the manuscript is poorly written and is confusing and doesn’t provide any clear picture.

Major issues:

1. The authors state the title of the manuscript as ‘Investigation of virulence factors…..”. The manuscript has little to do with an investigation of the virulence factors. This is a misleading title.

2. Same results are presented repeatedly with different percentages in different contexts. Like percent of MRSA and MSSA that were lukS/F-PV positive. It is then presented as among the lukS/F-PV how many of them are MRSA and how many of them are MSSA. It is unclear what the authors are trying to accomplish by stating same facts in two different ways. Some of these facts are also presented in Tables.

3. Lines 329-324: It is relatively accepted that HA-MRSA are resistant to more antibiotics compared to CA-MRSA isolates. Authors find a bit different results but no real explanation.

4. Authors also report much higher HA-MRSA strains to be lukS/F-PV positive. Is it possible that the authors are using a relatively loose metric to classify the isolates as CA-MRSA or HA-MRSA.

5. Authors repeatedly mention in the discussion section that their results are different than what has been reported in the literature but provide no comment on this except stating that the published study was based on small sample size.

6. The take home message of this manuscript is unclear.

Minor issues:

1. What is GERM-SA?

2. Line 115: Why not say from January 2013 to January 2016?

3. Line 117: Why the 21-day exclusion rules out duplicate isolation?

4. Lines 239-242: Very unclear the way it is stated. What does authors mean by ‘….considered to potentially harbor the lukS/F-PV gene?”

5. Lines 243-245: Why not simplify as “27 of 394 MSSA strains possessed lukS/F-PV while 81 of the 374 MSSA possessed this gene.”

Reviewer #2: Minor comments

Lines 133-139: I suggest this (Each isolate was plated onto a 5% blood agar plate (Diagnostic Media Products, National Health Laboratory Service, South Africa) followed by organism identification and antimicrobial susceptibility testing using automated systemsVITEK II system (bioMèrieux, Marcy l'Etoile, France)/MALDI-TOF MS (Microflex, Bruker Daltonics, MA, USA) and MicroScan Walkaway system (Siemens, Sacramento, CA, USA), (Gram-positive panel PM33) respectively).

Lines 152: You have to explain the screening methods of methicillin resistant briefly in the main text as you have shown in Fig 1. In the section bacterial isolates and phenotypic assays

Lines 164: You have to add in the supplementary section the primers of multiplex PCR assays.

Lines 412,430,426: Please correct your references.

6. PLOS authors have the option to publish the peer review history of their article (what does this mean?). If published, this will include your full peer review and any attached files.

Reviewer #1: No

Reviewer #2: No

---

## [Author Response · Author response to Decision Letter 0]

1 Oct 2019

Dear Editor

Thank you for considering the manuscript: “Investigation of virulence factors: SCCmec types and the Panton-Valentine Leukocidin exotoxin in South African blood culture Staphylococcus aureus surveillance isolates, 2013-2016” for publication. 

The authors have addressed the reviewer’s comments as follows: 

Reviewer 1

Major issues:

1. The authors state the title of the manuscript as ‘Investigation of virulence factors…..”. The manuscript has little to do with an investigation of the virulence factors. This is a misleading title.

The title has been changed to “Unconventional SCCmec types and low prevalence of the Panton-Valentine Leukocidin exotoxin in South African blood culture Staphylococcus aureus surveillance isolates, 2013-2016” which indicates the findings in this study.

2. Same results are presented repeatedly with different percentages in different contexts. Like percent of MRSA and MSSA that were lukS/F-PV positive. It is then presented as among the lukS/F-PV how many of them are MRSA and how many of them are MSSA. It is unclear what the authors are trying to accomplish by stating same facts in two different ways. Some of these facts are also presented in Tables.

The authors agree with the reviewer’s comment and have removed the statement showing the percentage of MRSA and MSSA that were selected for PVL-screening (line 240-241). Only the percentage of MRSA and MSSA in the PLV-positive group are shown (lines 246-247).

3. Lines 329-324: It is relatively accepted that HA-MRSA are resistant to more antibiotics compared to CA-MRSA isolates. Authors find a bit different results but no real explanation.

Although the findings from this study show that the type IV isolates were more resistant than type II isolates, a large number of these type IV isolates were indeed classified as HA-MRSA which is in keeping with other studies that showed that their HA strains were more resistant. This is mentioned in the latter part of the paragraph (lines 338-340). Differences in results could be attributed to clonal evolution and variations in different geographical locations – this has been included in lines 361-363.

4. Authors also report much higher HA-MRSA strains to be lukS/F-PV positive. Is it possible that the authors are using a relatively loose metric to classify the isolates as CA-MRSA or HA-MRSA.

The criteria used to classify CA- and HA-MRSA infections are described in the Methods section (Case Definition) and are acceptable criteria for defining these infections. This finding could be attributed to variations is different geographical locations (lines 361-363). However, it is also mentioned that there may be overlap of identical clones between these two groups (lines 305-307). This could potentially be another reason for this finding. 

5. Authors repeatedly mention in the discussion section that their results are different than what has been reported in the literature but provide no comment on this except stating that the published study was based on small sample size.

Possible explanations have been included (lines 320-330 and 361-363).

6. The take home message of this manuscript is unclear.

The last paragraph in the discussion addresses this – we have found that these unknown SCCmec types are largely unconventional and cannot be clearly defined. Additionally, the prevalence of PVL was low and more common in MSSA.

Minor issues:

1. What is GERM-SA?

This is a national surveillance platform at the National Institute for Communicable Diseases in South Africa. Previously, this was an acronym for the Group for Enteric, Respiratory and Meningeal Surveillance in South Africa. This platform was subsequently expanded to include other diseases such as those related to healthcare-associated infections. While GERMS-SA is still used, it is no longer referred to as the Group for Enteric, Respiratory and Meningeal Surveillance in South Africa. For this reason, it was not included in the manuscript. 

2. Line 115: Why not say from January 2013 to January 2016?

This has been corrected as suggested by the reviewer. 

3. Line 117: Why the 21-day exclusion rules out duplicate isolation?

A surveillance case definition of 21 days was applied to exclude duplicates i.e. to exclude infection caused by the same S. aureus strain. This is because if the patient acquired a subsequent blood stream infection within 21 days, the infection would probably be due to the same S. aureus strain. One would expect, that, provided adequate treatment was administered, the specific S. aureus infection would have been treated within this time-frame. Any infection with S. aureus after 21 days is considered to be a new infection. 

4. Lines 239-242: Very unclear the way it is stated. What does authors mean by ‘….considered to potentially harbor the lukS/F-PV gene?”

It is previously stated that PVL is linked to these infection types (line 59-62). Therefore no further explanation is required and this statement has been removed.

5. Lines 243-245: Why not simplify as “27 of 394 MSSA strains possessed lukS/F-PV while 81 of the 374 MSSA possessed this gene.”

 As mentioned in response to comment 2 (major issues), the statement showing the percentages and numbers of MRSA and MSSA that were selected for PVL-screening (line 240-241) have been removed and we have decided to focus on the PVL-positive group. For this reason, the denominator used is that of the PVL-positive isolates. 

Reviewer 2

Minor Issues:

1. Lines 133-139: I suggest this (Each isolate was plated onto a 5% blood agar plate (Diagnostic Media Products, National Health Laboratory Service, South Africa) followed by organism identification and antimicrobial susceptibility testing using automated systemsVITEK II system (bioMèrieux, Marcy l'Etoile, France)/MALDI-TOF MS (Microflex, Bruker Daltonics, MA, USA) and MicroScan Walkaway system (Siemens, Sacramento, CA, USA), (Gram-positive panel PM33) respectively).

This has been corrected as per the reviewer’s suggestion.

2. Lines 152: You have to explain the screening methods of methicillin resistant briefly in the main text as you have shown in Fig 1. In the section bacterial isolates and phenotypic assays

This has been included in the text as per the reviewer’s recommendation. 

3. Lines 164: You have to add in the supplementary section the primers of multiplex PCR assays.

As the method and primers used are well-known and published, the list of primers were not included. However, the method and primers have been referenced (line 170).

4. Lines 412,430,426: Please correct your references.

The references have been corrected as requested.

Moreover, the grammar and spelling throughout the manuscript have been examined and addressed.

Corresponding author: 

Dr. Ashika Singh-Moodley 

Senior Medical Scientist

Centre for Healthcare-associated infections, Antimicrobial Resistance and Mycoses

Antimicrobial Resistance Laboratory and Culture Collection 

1 Modderfontein Road, Sandringham

Private Bag X4, Sandringham, Johannesburg 2131

South Africa

Tel: +27 (0)11 555 0342 | Fax: +27 (0) 11 555 0430 

ashikas@nicd.ac.za / singh.ashika@gmail.com

I wish to thank you for the opportunity to re-submit this manuscript that investigates SCCmec types and the Panton-Valentine Leukocidin exotoxin in South African blood culture Staphylococcus aureus surveillance isolates and I hope that our attempts at addressing the comments are acceptable.

Kind regards 

Ashika Singh-Moodley

---

## [Decision Letter · Decision Letter 1]

12 Nov 2019

Unconventional SCCmec types and low prevalence of the Panton-Valentine Leukocidin exotoxin in South African blood culture Staphylococcus aureus surveillance isolates, 2013-2016.

PONE-D-19-23434R1

Dear Dr. Singh-Moodley, 

We are pleased to inform you that your manuscript has been judged scientifically suitable for publication and will be formally accepted for publication once it complies with all outstanding technical requirements.

With kind regards,

Davida S. Smyth, Ph.D.

Academic Editor

PLOS ONE

Additional Editor Comments (optional):

Reviewers' comments:

Reviewer's Responses to Questions

**Comments to the Author**

1. If the authors have adequately addressed your comments raised in a previous round of review and you feel that this manuscript is now acceptable for publication, you may indicate that here to bypass the “Comments to the Author” section, enter your conflict of interest statement in the “Confidential to Editor” section, and submit your "Accept" recommendation.

Reviewer #2: All comments have been addressed

2. Is the manuscript technically sound, and do the data support the conclusions?

Reviewer #2: Yes

3. Has the statistical analysis been performed appropriately and rigorously? 

Reviewer #2: Yes

4. Have the authors made all data underlying the findings in their manuscript fully available?

Reviewer #2: Yes

5. Is the manuscript presented in an intelligible fashion and written in standard English?

Reviewer #2: (No Response)

6. Review Comments to the Author

Reviewer #2: (No Response)

7. PLOS authors have the option to publish the peer review history of their article (what does this mean?). If published, this will include your full peer review and any attached files.

Reviewer #2: Yes: Dr Raspail Carrel Founou

---

## [Editor Report · Acceptance letter]

15 Nov 2019

PONE-D-19-23434R1 

Unconventional SCCmec types and low prevalence of the Panton-Valentine Leukocidin exotoxin in South African blood culture Staphylococcus aureus surveillance isolates, 2013-2016. 

Dear Dr. Singh-Moodley:

I am pleased to inform you that your manuscript has been deemed suitable for publication in PLOS ONE. Congratulations! Your manuscript is now with our production department. 

With kind regards,

on behalf of

Dr. Davida S. Smyth 

Academic Editor

PLOS ONE